# Proliferation-Based WHO Grading and Heterogeneous Gastrin Expression in Canine Gallbladder Neuroendocrine Tumors

**DOI:** 10.3390/vetsci12100989

**Published:** 2025-10-14

**Authors:** Yen-Tse Wu, Nadia Kelly, Ingeborg M. Langohr, Set Sokol, Jodie Gerdin, Chin-Chi Liu, Tyler J. Butsch, Andrea N. Johnston

**Affiliations:** 1Emergency & Critical Care, College of Veterinary Medicine, North Carolina State University, Raleigh, NC 27607, USA; ywu95@ncsu.edu; 2Department of Veterinary Clinical Sciences, School of Veterinary Medicine, Louisiana State University, Baton Rouge, LA 70803, USA; nkelly5@lsu.edu; 3Global Discovery Pathology & Multimodal Imaging, Translation Models Research Platform, Sanofi, Cambridge, MA 02141, USA; ingeborg.langohr@sanolfi.com; 4Department of Pathobiological Sciences, School of Veterinary Medicine, Louisiana State University, Baton Rouge, LA 70803, USA; 5Antech Diagnostics, Mars Petcare Science & Diagnostics, Loveland, CO 80538, USA; set.sokol@antechdx.com (S.S.); jodie.gerdin@antechmail.com (J.G.); 6Office of Research and Graduate Education, School of Veterinary Medicine, Louisiana State University, Baton Rouge, LA 70803, USA; cliu@lsu.edu; 7Departments of Pathobiology & Diagnostic Investigation and Small Animal Clinical Sciences, College of Veterinary Medicine, Michigan State University, East Lansing, MI 48824, USA; butschty@msu.edu

**Keywords:** cancer, carcinoid, dog, hepatobiliary, neuroendocrine neoplasm

## Abstract

Efforts by the World Health Organization (WHO) have clarified the descriptive nomenclature and histologic grading of neuroendocrine neoplasms (NENs) in human medicine. Employing a standardized stratification scheme enhances prognostic accuracy and guides treatment recommendations. Yet, this classification system has yet to be applied consistently in veterinary pathology. The objective of this study was to apply the markers of cell proliferation used in the WHO grading system to canine gallbladder (GB) NENs. In this group, all canine GB NENs were classified as low-grade neuroendocrine tumors (NETs), which is prognostically favorable in human patients. The secondary aim of the project was to determine whether canine GB NENs expressed gastrin, a potential therapeutic target. Gastrin expression was measured using immunohistochemistry; only 1 out of 19 GB NETs was positive. The use of proliferative indices in the histologic characterization of canine GB NENs is likely to improve prognostic information. The expression of gastrin was limited in this group of canine GB NENs; assessment may be warranted in individual canine NEN patients experiencing emesis or hematemesis.

## 1. Introduction

Gallbladder neuroendocrine neoplasms (GB NENs) are one of the rarest hepatobiliary cancers in dogs [1]. Brachycephalic breeds, especially Boston terriers, appear to be overrepresented [2]. As in people, it is likely that GB NENs arise from local multipotent stem cells, neuroendocrine cells of the biliary tract, or cells of neuroendocrine phenotype that may be observed in cases of chronic cholecystitis. Definitive diagnosis relies on histology, such as H&E-stained sections complemented by immunohistochemical staining for neuroendocrine markers (e.g., chromogranin A or synaptophysin). Positive staining for gastrin has also been described in 13 canine GB NENs [2]. Yet, primary gastrin-secreting tumors of the gallbladder are extremely rare in human patients [3]. The secretion of gastrin stimulates enterochromaffin-like cells to release histamine and directly promotes gastric acid production by parietal cells; gastrin secretion may correlate with the clinical signs of vomiting and hematemesis reported in dogs with GB NENs [2,4]. If this is a unique species-specific feature of canine GB NEN, gastrin expression could be targeted therapeutically to limit clinical signs and reduce tumor burden, as described in other gastrin-secreting canine neuroendocrine tumors [5,6,7,8,9,10].

NENs are a heterogeneous group of tumors that differ in morphology and biologic behavior [11,12]. The historical term ‘carcinoid’ fails to adequately describe the diversity of NENs. Properties including differentiation (grade), proliferation indices (Ki67%, mitoses), and clinical stage guide therapeutic recommendations and specific tumor antigen expression profiles have theranostic potential [13]. Guidelines put forward by The International Agency for Research on Cancer—World Health Organization (IARC-WHO) divided NENs into three broad categories based on histopathologic features: neuroendocrine tumors (NETs), neuroendocrine carcinomas (NECs), and mixed neuroendocrine and non-neuroendocrine neoplasms [14,15]. NETs are well-differentiated tumors and further subclassified into three grades: grade 1 (<2 mitoses/2 mm^2^ and/or Ki67 < 3%), grade 2 (2–20 mitoses/2 mm^2^ and/or Ki67 3–20%), and grade 3 (>20 mitoses/2 mm^2^ and/ or Ki67 > 20%). NETs have lower mitotic indices and are unlikely to metastasize NECs are poorly differentiated tumors with high mitotic indices (>20 mitoses/2 mm^2^ and/or Ki67 > 20%) and metastatic potential. Thus, NEN characterization and grading requires (1) the identification of neuroendocrine cells, (2) an assessment of cell differentiation, and (3) the measurement of cell proliferation, which has proven prognostically relevant in human patients. Yet, this classification system has yet to be widely applied in veterinary pathology. As highlighted in a recent retrospective review, the lack of standardized nomenclature and consistent histologic scoring applied to canine GB NENs hinders interspecies comparisons of tumor biology and prognostic guidance in regard to treatment response [1]. Therefore, the primary objective of this study was to use the WHO classification system to grade an archival set of canine GB NENs. The secondary aim of the project was to determine whether canine GB NENs expressed gastrin, a potential therapeutic target.

## 2. Materials and Methods

### 2.1. Sample Identification

The Antech Diagnostics (Mars Petcare Science & Diagnostics, Loveland, CO, USA) database was searched for canine GB NEN biopsies (including oncology and hepatopathy specialty cases) using the keywords liver AND neuroendocrine, gallbladder AND neuroendocrine, gall bladder AND neuroendocrine, liver AND carcinoid, gallbladder AND carcinoid, and gall bladder AND carcinoid. A 3-year date range (October 2018 through October 2021) was searched. The word liver was included to capture miscoded samples. Inclusion criteria included an unambiguous diagnosis of a neuroendocrine tumor primary to the liver or gallbladder and canine species. Samples were excluded if insufficient tissue was available for additional testing (no needle biopsies) or if the NEN was from in the liver and not the GB. All samples were from the United States. H&E-stained slides from formalin-fixed, paraffin-embedded (FFPE) canine GB NENs were retrospectively reviewed by a board-certified anatomic pathologist (S.S.). The search identified 20 canine GB NEN cases. Histologic characteristics and de-identified patient data, including age and sex, were recorded.

### 2.2. Histology and Immunohistology

Gastrin immunohistochemical labeling (Dako A0568, 1:200; Agilent Technologies, Carpinteria, CA, USA) was performed by the Animal Health Diagnostic Center at the Cornell University College of Veterinary Medicine; this antibody has been utilized by multiple researchers to identify gastrin secretion in dogs [16,17,18,19,20]. In brief, sections were dewaxed (Bond Dewax Solution, Leica, Teaneck, NJ, USA) and processed for heat epitope retrieval (Bond Epitope Retrieval solution, Leica) for 30 min, peroxide block (Leica) was applied for 5 min, followed by incubation with the primary antibody for 30 min. Next, a polymer (Leica Bond Polymer Refine Detection, Leica) was applied for 10 min followed by DAB (Leica Bond Polymer Refine Detection, Leica) for 10 min, and hematoxylin (Leica) counterstain for 5 min. A positive control reference tissue (canine gastric wall) was included for comparison (Appendix A). Synaptophysin immunostaining (Abcam ab14692, 1:100; Abcam, Cambridge, MA, USA) and Ki67 (Dako clone MIB-1 M7240, 1:100) detection were performed by the Animal Health Diagnostic Center at the Cornell University College of Veterinary Medicine as previously described [21,22]. Immunohistochemical labeling with synaptophysin (Syn), mitotic count, and Ki67% were determined by a board-certified anatomic pathologist (I.L.). In brief, mitotic cell counts were quantified per 2.37 mm^2^ (10 high power fields at 400× magnification). Classification categories included well differentiated NET, grade 1 (< 2 mitoses/2 mm^2^ and/or Ki67 < 3%); NET, grade 2 (2–20 mitoses/2 mm^2^ and/or Ki67 3–20%); NET, grade 3 (>20 mitoses/2 mm^2^ and/or Ki67 >20%); and poorly differentiated NECs (>20 mitoses/2 mm^2^ and/or Ki67 >20% (often >70%). Quantitative assessment of Ki67 immunostaining was conducted using the Cytonuclear v2 algorithm on the HALO® whole-slide image analysis platform (Indica Labs, Albuquerque, NM, USA). The analysis focused on viable tumor area hot spots, with necrotic regions and lymphocytic aggregates manually excluded. Results were reported as a percentage of immunopositive cells (Ki67%). 

### 2.3. Statistical Analysis

Statistical analyses were performed using GraphPad Prism, version 9.4 (GraphPad Prism Software, La Jolla, CA, USA). Data are expressed as median and range. A Pearson correlation with linear regression was performed on proliferation indices. Significance was set at *p* < 0.05.

## 3. Results

### 3.1. Patient Demographics

Patient age and sex was available for 19 out of 20 dogs. The median patient age was 10 years (range: 7–13 years). There was one intact female, three unaltered males, eight spayed females, and seven castrated males. 

### 3.2. Histologic Characteristics

All GB NENs showed typical histological characteristics of NENs, consisting of nests and packets of round to polygonal cells within a fine fibrovascular stroma and slightly granular basophilic cytoplasm [1,23]. All 20 samples were positive for synaptophysin immunostaining, whereas only one sample was weakly positive for gastrin immunostaining (Figure 1). Additional descriptive features included coagulative nerosis (*n* = 13), peripheral invasion (17), acute hemorrhage or blood filled spaces (11), hemosiderin deposits (6), and large or irregular nuclei (10, Appendix A). Ki67% was assessed in 18 cases while two samples were excluded due to poor tissue preservation or insufficient tissue area for complete analysis. Median Ki67% was 1.2% (range: 0.22–3.7%). Mitotic count was assessed in 20 samples. Based on the two proliferative indices, all 18 samples were classified as NETs. One sample met the criteria for grade 2 (Ki67 3.7%), whereas the remaining 17 samples were classified as grade 1. Median mitotic count was 3.5 cells per 2.37 mm^2^ (range: 0–13). Ki67% and mitotic count had a significant positive correlation (r = 0.4993, *p* = 0.0349; Figure 2).

## 4. Discussion

Using the guidelines set forth by the WHO, the dogs in this group had uniformly low-grade NETs. In dogs with GB NENs, mean survival times exceeding three years have been reported following cholecystectomy, which is consistent with reports of survival in human patients with completely excised grade 1 NETs [1,16]. These results highlight how the application of the WHO guidelines to canine GB NETs may provide valuable prognostic information in our canine patients. There is moderate correlation between mitotic count and Ki67%, yet Ki67% may enhance grading accuracy. Uniformity across classification systems will simplify correlations in prospective survival studies following therapeutic intervention such as somatostatin or gastrin inhibition.

The limited expression of gastrin in this canine group compared to previous studies demonstrates the heterogeneity of GB NENs [2,16]. While general screening and treatment for gastrin overexpression (e.g., netazepide) in canine NENs is unwarranted, it may be considered an ancillary test/therapeutic in specific cases with characteristic clinical signs of hypergastrinemia such as vomiting and hematemesis or in dogs with definitive NECs [2,5,6]. Further investigation into somatostatin receptor expression is needed in dogs with NENs. Physiologically, somatostatin is an endogenous inhibitory hormone that represses the secretion of gastrin [6,11,24,25]. Somatostatin analogs (SSAs) are commonly used in human NEN patients as a first-line therapy [26]. Octreotide, the somatostatin analog, is well tolerated in dogs and could be considered as an option to limit the clinical signs associated with gastrin expression from GB NENs [27].

The authors acknowledge that the major limitations of this study are the lack of associated clinical information and small sample size. Even from the limited demographic data available, the sex distribution of dogs in this group yielded an interesting finding. Unlike previously reported canine GB NEN cases, a male sex predilection was not demonstrated herein [1,2,16]. This may be attributed to the impact of ovariohysterectomy and castration, as the majority of dogs in this group were sexually altered; however, this has not been rigorously investigated as a unique biological variable. Future studies would be warranted to combine clinical picture and histopathologic information. The use of alternative anti-gastrin antibodies and IHC protocols should also be explored to clarify the differing results in this study versus the findings of O’Brien, et al.

## 5. Conclusions

The use of a standardized grading system in the histological characterization of canine GB NENs is likely to improve clinical prognostic information. Given the heterogeneous expression of gastrin in canine GB NENs, it is unlikely to be widely applicable as a druggable target but may be relevant in a subset of dogs with NENs.

## Figures and Tables

**Figure 1 vetsci-12-00989-f001:**
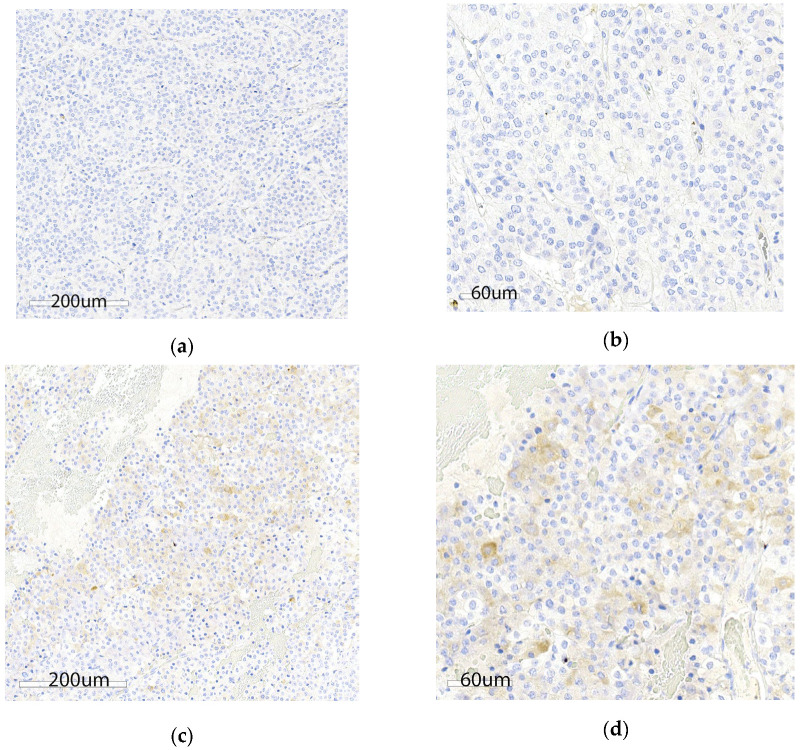
Gastrin immunohistochemistry of formalin-fixed paraffin-embedded gallbladder neuroendocrine neoplasms. (**a**) Low magnification (20×) of a gastrin immunonegative sample. (**b**) High magnification (40×) of a gastrin immunonegative sample. (**c**) Low magnification (20×) of the gastrin immunopositive sample, wherein ~90% of neoplastic cells are positive. Positive cells display weak to moderate cytoplasmic brown stain (DAB with hematoxylin counterstain). (**d**) High magnification (40×) of a gastrin immunopositive sample. Positive cells display weak-to-moderate cytoplasmic brown stain (DAB counterstain).

**Figure 2 vetsci-12-00989-f002:**
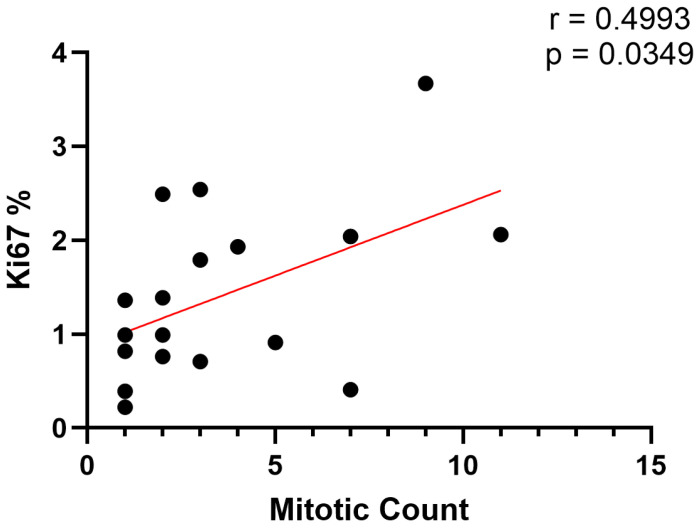
Correlation between Ki67% and mitotic count. The Ki67% are displayed on the y-axis and mitotic count values are displayed on the x-axis for *n* = 18 dogs. Each black dot represents 1 dog. The linear trend of the data is depicted by the red line.

## Data Availability

The original contributions presented in this study are included in the article/Appendix A. Further inquiries can be directed to the corresponding author.

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
