# Peer review of "Proliferation-Based WHO Grading and Heterogeneous Gastrin Expression in Canine Gallbladder Neuroendocrine Tumors"

_vetsci, 2025, doi:10.3390/vetsci12100989_

Round 1

Reviewer 1 Report

Comments and Suggestions for Authors

This study aimed to apply the WHO classification system on neuroendocrine neoplasms to canine neuroendocrine gallbladder neoplasms. A secondary aim was to determine gastrin expression in GB NENs, although this secondary aim was not justified. Overall, the study is poorly written, the methodology is flawed and hence this study can’t be published in its current form. The authors concluded that the lack of gastrin expression in their GB NEN cohort is simply due to differences between the current cohort and the cohort of O’Brien et al., (2021) who demonstrated moderate to strong gastrin expression with immunohistochemistry (IHC) in all GB NENs (n = 13) in their study. O’Brien et al. provided a detailed protocol including stomach as positive control for their gastrin IHC, whereas in the current study there is no mention of positive and negative controls and the gastrin antibody has not even been disclosed. Hence, the lack of gastrin staining seems more likely to be a methodological issue. Please find my additional comments below:

Introduction

Lines 68-69: it states that differentiation (grade), proliferation indices, and clinical stage are required for diagnosis. In the narrowest sense of the definition of ‘diagnosis’ this is not correct. For example, clinical stage is not required for the diagnosis. According to the WHO 2022 classification diagnostic criteria for GB NENs are: neuroendocrine morphology on histopathology, diffuse and intense expression of cytokeratin(s) and chromogranin A or two other NE markers; Ki-67.

Lines 74-76: authors seem to apply that NETs can only be grade 1 or 2 and grade 3 tumours are per definition NECs. This is not true. NECs are always grade 3, but NETs can be grade 1, 2, or 3.

Line 84-85: more detailed information on gastrin expression should be provided in the introduction. Why did the authors set out the study gastrin IHC again, what reasons did they have to doubt the gastrin results from the O’Brien et al., publication?

Materials and methods

Line 94: although the studies seems to imply that only GB NENs were included, it states here that both liver and GB NENs were included. What is correct? Please revise accordingly.

Line 98: 20 cases are mentioned. The rest of the study seems to imply that there were only 19 cases. Please amend.

Line 103-104: the authors refer to reference 9 for IHC protocols. This reference however does not provide detailed information on protocols and antibodies used. Please provide antibodies, dilutions, and essential steps from the IHC protocols (e.g. antigen retrieval, incubation times). Also please provide information on negative and positive controls and as a minimum include a picture of the gastrin positive control.

Line 120: p should be written in italics.

Results

Lines 124-125: values lower than 10 should be written in full

Line 124: ‘unaltered’ should read ‘intact’.

Lines 129/136: please revise 20 samples to correct number.

Line 132: n should be written in italics. Please add n to all case numbers in brackets.

Line 140: p should be written in italics.

Figure 1: please add the size of the scale bar to the figure legend.

Discussion

All comments regarding value of the findings in terms of prognostic information should be removed, because no such conclusions can be made from this study due to the lack of clinical information. Although the authors did not have access to follow-up outcome data, could the authors perhaps add information regarding clinical stages of disease?

There is also no point in discussing therapeutic intervention targeting gastrin if the results from this study would truly reflect that GB NENs do not express gastrin (which I do not believe to be true).

Author Response

Comment 1: Comments and Suggestions for Authors

This study aimed to apply the WHO classification system on neuroendocrine neoplasms to canine neuroendocrine gallbladder neoplasms. A secondary aim was to determine gastrin expression in GB NENs, although this secondary aim was not justified. Overall, the study is poorly written, the methodology is flawed and hence this study can’t be published in its current form. The authors concluded that the lack of gastrin expression in their GB NEN cohort is simply due to differences between the current cohort and the cohort of O’Brien et al., (2021) who demonstrated moderate to strong gastrin expression with immunohistochemistry (IHC) in all GB NENs (n = 13) in their study. O’Brien et al. provided a detailed protocol including stomach as positive control for their gastrin IHC, whereas in the current study there is no mention of positive and negative controls and the gastrin antibody has not even been disclosed. Hence, the lack of gastrin staining seems more likely to be a methodological issue. Please find my additional comments below:

 Response 1: Thank you for taking the time to review the manuscript. The authors have responded to the specific points of revision below.

Comment 2: Introduction

Lines 68-69: it states that differentiation (grade), proliferation indices, and clinical stage are required for diagnosis. In the narrowest sense of the definition of ‘diagnosis’ this is not correct. For example, clinical stage is not required for the diagnosis. According to the WHO 2022 classification diagnostic criteria for GB NENs are: neuroendocrine morphology on histopathology, diffuse and intense expression of cytokeratin(s) and chromogranin A or two other NE markers; Ki-67.

Response 2: The authors apologize for the confusion. In lines 68-69, “diagnosis” is being used from the perspective of a clinician not a histopathologist. As stated in lines 59-60, definitive diagnosis relies on histology and immunohistochemical markers.  To address the reviewers’ concern, the line has been changed to the following: “Properties including differentiation (grade), proliferation indices (Ki67 %, mitoses), and clinical stage guide therapeutic recommendations and specific tumor antigen….”.

Comment 3: Lines 74-76: authors seem to apply that NETs can only be grade 1 or 2 and grade 3 tumours are per definition NECs. This is not true. NECs are always grade 3, but NETs can be grade 1, 2, or 3.

Response 3: Thank you for noting this. The introduction has been amended to clarify the WHO classification (lines 72-80): “NENs into three broad categories based on histopathologic features: neuroendocrine tumors (NETs), neuroendocrine carcinomas (NECs), and mixed neuroendocrine and non-neuroendocrine neoplasms.[7,8] NETs are well-differentiated tumors are further subclassified into three grades: grade 1 (< 2 mitoses/ 2 mm2 and/or Ki67 < 3%), grade 2 (2 - 20 mitoses/ 2 mm2 and/ or Ki67 3- 20%), and grade 3 (> 20 mitoses/ 2 mm2 and/ or Ki67 > 20%). NETs have lower mitotic indices and are less likely to metastasize. NECs are poorly differentiated tumors with high mitotic indices (> 20 mitoses/ 2 mm2 and/ or Ki67 > 20%) and metastatic potential.”

Comment 4”: Line 84-85: more detailed information on gastrin expression should be provided in the introduction. Why did the authors set out the study gastrin IHC again, what reasons did they have to doubt the gastrin results from the O’Brien et al., publication?

 Response 4: Experimental findings should always be validated by multiple studies, particularly when the protein expressed could be a therapeutic target. The following line has been added to the introduction: “It is possible that gastrin expression could be targeted therapeutically.”

Comment 5: Materials and methods

Line 94: although the studies seems to imply that only GB NENs were included, it states here that both liver and GB NENs were included. What is correct? Please revise accordingly.

Response 5: Only GB NEN were included in the study. The search term “liver” was included to capture miscoded samples. The exclusion criteria have been amended to included “NEN was from the the liver and not the GB” (line 101) were excluded.

Comment 6: Line 98: 20 cases are mentioned. The rest of the study seems to imply that there were only 19 cases. Please amend.

Response 6: Twenty samples were collected and included in the study. Not all data were available for 20 samples. The exclusions are stated in the text (line 138). Under patient demographics, “out of 20” has been added to line 128.

Comment 7:  Line 103-104: the authors refer to reference 9 for IHC protocols. This reference however does not provide detailed information on protocols and antibodies used. Please provide antibodies, dilutions, and essential steps from the IHC protocols (e.g. antigen retrieval, incubation times). Also please provide information on negative and positive controls and as a minimum include a picture of the gastrin positive control.

Response 7: Thank you for pointing this out. The specific IHC protocol and antibodies used have been added to the methods (lines 107-115).

Comment 8: Line 120: p should be written in italics.

Response 8: The letter has been italicized.

Comment 9: Results

Lines 124-125: values lower than 10 should be written in full

Response 9: Values lower than 10 have been written “in full”.

Comment 10: Line 124: ‘unaltered’ should read ‘intact’.

Response 10: The requested change has been completed.

Comment 11: Lines 129/136: please revise 20 samples to correct number.

Response 11: Twenty is the correct number.

Comment 12: Line 132: n should be written in italics. Please add n to all case numbers in brackets.

Response 12: The n has been italicized. The numbers of samples are included in the text.

Comment 13: ” Line 140: p should be written in italics.

Response 13: The p has been italicized.

Comment 14: Figure 1: please add the size of the scale bar to the figure legend.

Response 14: The size of the scale bar is written on the scale bar in the images.

Commentv15: Discussion

All comments regarding value of the findings in terms of prognostic information should be removed, because no such conclusions can be made from this study due to the lack of clinical information. Although the authors did not have access to follow-up outcome data, could the authors perhaps add information regarding clinical stages of disease?

Response 15: Proliferation indices unequivocally correlate with tumor growth. Clinically, how quickly a tumor increases in size is prognostically important. The authors fail to see the relevance of removing a discussion of the prognostic potential of proliferative indices. Sadly, data on stage were not available.

Comment 16: There is also no point in discussing therapeutic intervention targeting gastrin if the results from this study would truly reflect that GB NENs do not express gastrin (which I do not believe to be true).

Response 16: The authors respectfully disagree as some, but not all, GB NENS do express gastrin. If the reviewer “believes” that the gastrin is expressed in all GB NENs, then this should be of value to him or her as well.

Reviewer 2 Report

Comments and Suggestions for Authors

In dogs, gallbladder neuroendocrine tumors (GB-NENs) are a rare type of cancer, originating from neuroendocrine cells within the gallbladder. These tumors are uncommon, but their occurrence is increasingly recognized due to advancements in diagnostic imaging. While they can arise in the hepatobiliary system, gallbladder carcinoids are particularly rare. While there are very few reports of the available data in the literature, the manuscript needs major changes/details implements.

See below the major concerns that could improve it.

  Abstract 

I consider the cohort definition to be a methodological error. In statistics, a cohort is a group of individuals who share a common characteristic or experience within a defined time, related to time (like birth year) or other factors (a group of patients starting a new medication at the same time). In summary, a cohort study, in turn, is an observational study that follows such a group over time to examine how certain factors influence their health or other outcomes.   

Introduction

A key characteristic enabling the identification of neuroendocrine cells is the identification of broad-spectrum markers which include at least Chromogranin, Synaptophysin and NSE.

The guidelines mentioned in the paper include 3 broad categories based on histopathology: well differentiated NETs, poorly differentiated NECs and mixed neuroendocrine carcinoma which have features of NENs and non-neuroendocrin neoplasms (MiNENs). NETs are further classified in grade 1, 2 and 3 based on mitotic and Ki67 index.

Material and methods

Information on clinical data is missing, and breeds are not reported (potential risk factor).

It follows that the results and discussion appear unsatisfactory based on the findings made.

The bibliography is updated and appropriate.

Author Response

Comments and Suggestions for Authors

In dogs, gallbladder neuroendocrine tumors (GB-NENs) are a rare type of cancer, originating from neuroendocrine cells within the gallbladder. These tumors are uncommon, but their occurrence is increasingly recognized due to advancements in diagnostic imaging. While they can arise in the hepatobiliary system, gallbladder carcinoids are particularly rare. While there are very few reports of the available data in the literature, the manuscript needs major changes/details implements.

See below the major concerns that could improve it.

Response:

Thank you for your thoughtful review.

Comment 1:  Abstract 

I consider the cohort definition to be a methodological error. In statistics, a cohort is a group of individuals who share a common characteristic or experience within a defined time, related to time (like birth year) or other factors (a group of patients starting a new medication at the same time). In summary, a cohort study, in turn, is an observational study that follows such a group over time to examine how certain factors influence their health or other outcomes.  

Response 1:

“Cohort” has been replaced with “group”. 

Comment 2: Introduction

A key characteristic enabling the identification of neuroendocrine cells is the identification of broad-spectrum markers which include at least Chromogranin, Synaptophysin and NSE.

The guidelines mentioned in the paper include 3 broad categories based on histopathology: well differentiated NETs, poorly differentiated NECs and mixed neuroendocrine carcinoma which have features of NENs and non-neuroendocrin neoplasms (MiNENs). NETs are further classified in grade 1, 2 and 3 based on mitotic and Ki67 index.

Response 2:

All samples stained positive for synaptophysin (lines: 106-109).

Thank you to the reviewer for noting this; it was very confusing as written. The introduction has been amended to include MiNENs and to clarify the WHO classification (lines 72-80): “NENs into three broad categories based on histopathologic features: neuroendocrine tumors (NETs), neuroendocrine carcinomas (NECs), and mixed neuroendocrine and non-neuroendocrine neoplasms.[7,8] NETs are well-differentiated tumors are further subclassified into three grades: grade 1 (< 2 mitoses/ 2 mm2 and/or Ki67 < 3%), grade 2 (2 - 20 mitoses/ 2 mm2 and/ or Ki67 3- 20%), and grade 3 (> 20 mitoses/ 2 mm2 and/ or Ki67 > 20%). NETs have lower mitotic indices and are less likely to metastasize. NECs are poorly differentiated tumors with high mitotic indices (> 20 mitoses/ 2 mm2 and/ or Ki67 > 20%) and metastatic potential.”

Comment 3: Material and methods

Information on clinical data is missing, and breeds are not reported (potential risk factor). It follows that the results and discussion appear unsatisfactory based on the findings made.

Response 3:

The authors agree with the reviewer that breed information would be interesting. Unfortunately, as stated in the manuscript, breed was not included in the de-identified patient description provided by our commercial partner. Nevertheless, patient information is not required for the histologic sample review criteria described in the manuscript and, although brachycephalic breeds are over-represented in previous studies, currently risk factors are unknown for canine gallbladder neuroendocrine neoplasms.

Comment 4:  The bibliography is updated and appropriate.

Response 4:

Thank you.

Round 2

Reviewer 1 Report

Comments and Suggestions for Authors

Thank you for addressing some of my previous concerns, which have improved the quality of your manuscript. Regrettably the manuscript still suffers from major flaws and my recommendation to reject this for publication in Veterinary Sciences remains the same.

Introduction:

Lines 65-66: the authors have added a sentence stating that gastrin expression could be targeted therapeutically. This statement has not been referenced and no further explanation is provided. 

Lines 78-80: ‘NETs have lower mitotic indices are unlikely to metastasize.’ ‘and’ appears to be missing in this sentence. Also, what comparison is made here: NETs vs NECs? This statement is too generalised, because grade 3 NETs and NECs both have mitotic indices >20 mitoses/mm2 and/or Ki67 >20%.

Lines 87-89: The introduction does not clearly articulate why applying the WHO grading system to canine GB NENs is necessary or novel. While it mentions that the system is not widely used in veterinary pathology, it fails to explain what limitations exist in current veterinary grading practices or how this study addresses them. The introduction heavily references human WHO classification and treatment paradigms but does not critically assess whether these are appropriate or validated for canine pathology. There is no discussion of species-specific differences in tumor biology, clinical presentation, or treatment response, which is essential when adapting human frameworks to veterinary medicine.

Lines 89-90: As previously stated, the rationale for investigating gastrin expression is weak. The introduction suggests therapeutic relevance but does not provide sufficient background on its prevalence or clinical impact in dogs.

Lines 98-101: It is very confusing that initially liver or gallbladder NENs were included, whereas liver NENs were subsequently excluded. From the rebuttal I understand that this approach was used to capture miscoded samples - however this should explicitly be explained in the body of the manuscript to avoid confusion.

Lines 107-116: Following my previous comment, the authors have added additional information regarding their IHC protocol. It is unclear what protocol they have added, presumably gastrin. However, all protocols should be provided including positive and negative controls. I also previously requested dilutions of antibodies used, this information is still missing. The authors also failed to include a picture of their positive gastrin control, this is still required as well. Furthermore, it appears a polyclonal rabbit anti-gastrin antibody was used which to the reviewer’s knowledge has not previously been validated for use on canine tissues. The authors should provide data supporting specificity and cross-reactivity of this antibody with canine tissue. Specificity can be demonstrated using a Western blot of canine tissue lysate: a band at the expected molecular weight for gastrin, lost on peptide blocking, confirms specificity.

Lines 139 - 152: several histopathologic criteria are presented here, including the number of cases that demonstrated specific parameters. From these data one can’t distill the combination of parameters per case. It would be helpful to include a table with all 18 samples detailing the different histopathologic variables per case.

Discussion

The discussion draws strong conclusions from a very small dataset (n = 20), including only one gastrin-positive case. Statements such as “application of the WHO guidelines may provide valuable prognostic information” are speculative and not supported by survival or outcome data in this study.

Despite only one tumor expressing gastrin, the discussion includes an extended section on gastrin-targeted therapy, which is disproportionate and misleading. The authors fail to critically assess why their findings contradict previous reports of gastrin expression and what this means for future research. 

The discussion heavily references human medicine (e.g., netazepide therapy, somatostatin analogs) without critically evaluating whether these approaches are relevant or feasible in veterinary practice.

The claim that Ki67 may not be required in all cases is based on a moderate correlation (r = 0.4993) and lacks statistical depth.

Author Response

Comment 2_1: Thank you for addressing some of my previous concerns, which have improved the quality of your manuscript. Regrettably the manuscript still suffers from major flaws and my recommendation to reject this for publication in Veterinary Sciences remains the same.

Response 2_1: The authors thank the reviewer for spending his or her time re-reviewing the manuscript and providing additional points of revision.

Introduction:

Comment 2_1: Lines 65-66: the authors have added a sentence stating that gastrin expression could be targeted therapeutically. This statement has not been referenced and no further explanation is provided. 

Response 2_1:  This line was added in response to the reviewer’s comment in the first review questioning why the study was undertaken (see [“”] below). Lines 61-69: References citing therapeutic targeting have been added to the text. Additionally, the introduction has been expanded further to address the discordance of O’Brien et al’s  findings with the human species and why, if validated, gastrin expression could be important therapeutically in dogs.

[“Comment 3: Line 84-85: more detailed information on gastrin expression should be provided in the introduction. Why did the authors set out the study gastrin IHC again, what reasons did they have to doubt the gastrin results from the O’Brien et al., publication?

 Response 3: Experimental findings should always be validated by multiple studies, particularly when the protein expressed could be a therapeutic target. The following line has been added to the introduction: “It is possible that gastrin expression could be targeted therapeutically.”]

Comment 2_2: Lines 78-80: ‘NETs have lower mitotic indices are unlikely to metastasize.’ ‘and’ appears to be missing in this sentence. Also, what comparison is made here: NETs vs NECs? This statement is too generalised, because grade 3 NETs and NECs both have mitotic indices >20 mitoses/mm2 and/or Ki67 >20%.

Response 2_2: The word “and” has been added to the sentence. As per the WHO grading system, grade 3 NETs and NECS have the same replicative indices. The difference, as stated in the text, is the degree of differentiation (lines 79 and 83-84).

Comment 2_3: Lines 87-89: The introduction does not clearly articulate why applying the WHO grading system to canine GB NENs is necessary or novel. While it mentions that the system is not widely used in veterinary pathology, it fails to explain what limitations exist in current veterinary grading practices or how this study addresses them. The introduction heavily references human WHO classification and treatment paradigms but does not critically assess whether these are appropriate or validated for canine pathology. There is no discussion of species-specific differences in tumor biology, clinical presentation, or treatment response, which is essential when adapting human frameworks to veterinary medicine.

Response 2_3: The following text has been modified to address the reviewer’s concerns (lines 89-92): “As highlighted in a recent retrospective review, the lack of standardized nomenclature and consistent histologic scoring applied to canine GB NENs hinders interspecies comparisons of tumor biology and prognostic guidance in regard to treatment response.”

Comment 2_4: Lines 89-90: As previously stated, the rationale for investigating gastrin expression is weak. The introduction suggests therapeutic relevance but does not provide sufficient background on its prevalence or clinical impact in dogs.

Response 2_4: The introduction has been expanded to address this concern (below). Further, the discussion includes a detailed discussion of gastrin targeting.

Lines 61-69: “Positive staining for gastrin has also been described in 13 canine GB NENs.[2] Yet, primary gastrin secreting tumors of the gallbladder are extremely rare in human patients.[3] Secretion of gastrin stimulates enterochromaffin-like cells to release histamine and directly promotes gastric acid production by parietal cells; gastrin secretion may correlate with the clinical signs of vomiting and hematemesis reported in dogs with GB NENs.[2,4] If this is a unique species specific feature of canine GB NEN, gastrin expression could be targeted therapeutically to limit clinical signs and reduce tumor burden.[5,6]”

Comment 2_5: Lines 98-101: It is very confusing that initially liver or gallbladder NENs were included, whereas liver NENs were subsequently excluded. From the rebuttal I understand that this approach was used to capture miscoded samples - however this should explicitly be explained in the body of the manuscript to avoid confusion.

Response 2_5: The following sentence has been added to explain the use of the word “liver”: (Line 103) “The word liver was included to capture miscoded samples.”

Comment 2_6: Lines 107-116: Following my previous comment, the authors have added additional information regarding their IHC protocol. It is unclear what protocol they have added, presumably gastrin. However, all protocols should be provided including positive and negative controls. I also previously requested dilutions of antibodies used, this information is still missing. The authors also failed to include a picture of their positive gastrin control, this is still required as well. Furthermore, it appears a polyclonal rabbit anti-gastrin antibody was used which to the reviewer’s knowledge has not previously been validated for use on canine tissues. The authors should provide data supporting specificity and cross-reactivity of this antibody with canine tissue. Specificity can be demonstrated using a Western blot of canine tissue lysate: a band at the expected molecular weight for gastrin, lost on peptide blocking, confirms specificity.

Response 2_6: The authors apologize for excluding the concentration and positive gastrin antibody control; these errors have been corrected (1:200, see Figure S1). A thorough literature search has identified use of the Dako A0568 polyclonal rabbit gastrin antibody on canine tissue and tissue from other species by many researchers. Of note is the study by Lessels et al., which uses the Dako A0568 antibody as the validated positive control antibody to which an in-house antibody is compared. These studies have been cited, as have the protocols for all the protocols performed at Cornell AHDC laboratory.

Comment 2_7: Lines 139 - 152: several histopathologic criteria are presented here, including the number of cases that demonstrated specific parameters. From these data one can’t distill the combination of parameters per case. It would be helpful to include a table with all 18 samples detailing the different histopathologic variables per case.

Response 7: A supplemental table has been added with the specific histologic parameters per case (Table S1).

Discussion

Comment 2_7: The discussion draws strong conclusions from a very small dataset (n = 20), including only one gastrin-positive case. Statements such as “application of the WHO guidelines may provide valuable prognostic information” are speculative and not supported by survival or outcome data in this study.

Response 8: The dogs in this study have low grade NENs. This is highly relevant to prognosis and clinical decision-making regarding on-going treatment. This statement is relevant as documented by the study.

Comment 2_8: Despite only one tumor expressing gastrin, the discussion includes an extended section on gastrin-targeted therapy, which is disproportionate and misleading. The authors fail to critically assess why their findings contradict previous reports of gastrin expression and what this means for future research. 

Response 2_8: The authors respectfully disagree. Based on the data from the O’Brien, et al manuscript and for the single dog in this study with gastrin expression, a subset of dogs are likely to benefit from targeted therapy. Given the small sample numbers (n=13 in O’Brien, et al and n=20 dogs herein) tumor heterogeneity, as mentioned in the discussion, is a reasonable assumption for the differing results.   Never-the-less, the following statement has been added to the discussion section: “Use of alternative anti-gastrin antibodies and IHC protocols should also be explored to clarify the differing results in this study versus the findings of O’Brien et al.”

Comment 2_9: The discussion heavily references human medicine (e.g., netazepide therapy, somatostatin analogs) without critically evaluating whether these approaches are relevant or feasible in veterinary practice.

Response 2_9: Human drugs are regularly used in veterinary patients. The somatostatin analogue octreotide is a treatment option in dogs with refractory protein losing enteropathy, which demonstrates feasibility. The following statement has been added to the discussion and referenced (lines 201-203): “Octreotide, the somatostatin analog, is well tolerated in dogs and could be considered as an option to limit clinical signs associated with gastrin expression from GB NEN.” Please see response 2_8 regarding the relevance of the statement.

Comment 2_10: The claim that Ki67 may not be required in all cases is based on a moderate correlation (r = 0.4993) and lacks statistical depth.

Response 2_10: This statement has been removed.

Round 3

Reviewer 1 Report

Comments and Suggestions for Authors

Thank you for addressing my previous comments